# Strategies for Overcoming Immune Evasion in Bladder Cancer

**DOI:** 10.3390/ijms25063105

**Published:** 2024-03-07

**Authors:** Juhyun Shin, Jeong Won Park, Seon Young Kim, Jun Ho Lee, Wahn Soo Choi, Hyuk Soon Kim

**Affiliations:** 1Department of Health Sciences, The Graduate School of Dong-A University, Busan 49315, Republic of Korea; juhyeun2883@naver.com (J.S.); pjwx096@naver.com (J.W.P.); tjsduddlmm@naver.com (S.Y.K.); 2Department of Korean Medicine, College of Korean Medicine, Woosuk University, Jeonju 54986, Republic of Korea; celtece@daum.net; 3Department of Immunology, College of Medicine, Konkuk University, Chungju 27478, Republic of Korea; wahnchoi@kku.ac.kr

**Keywords:** bladder cancer, immune evasion, immune checkpoint inhibitor, immunoregulatory cell, tumor microenvironment

## Abstract

Tumors intricately shape a highly immunosuppressive microenvironment, hampering effective antitumor immune responses through diverse mechanisms. Consequently, achieving optimal efficacy in cancer immunotherapy necessitates the reorganization of the tumor microenvironment and restoration of immune responses. Bladder cancer, ranking as the second most prevalent malignant tumor of the urinary tract, presents a formidable challenge. Immunotherapeutic interventions including intravesical BCG and immune checkpoint inhibitors such as atezolizumab, avelumab, and pembrolizumab have been implemented. However, a substantial unmet need persists as a majority of bladder cancer patients across all stages do not respond adequately to immunotherapy. Bladder cancer establishes a microenvironment that can actively hinder an efficient anti-tumor immune response. A deeper understanding of immune evasion mechanisms in bladder cancer will aid in suppressing recurrence and identifying viable therapeutic targets. This review seeks to elucidate mechanisms of immune evasion specific to bladder cancer and explore novel pathways and molecular targets that might circumvent resistance to immunotherapy.

## 1. Introduction

Bladder cancer is one of the most prevalent and malignant tumors that originates in cells within the bladder. In 2020, it ranked twelfth globally among diagnosed malignancies [1]. In its initial stages, bladder cancer may remain asymptomatic, while certain individuals may later experience symptoms such as urinary pain, the presence of blood in their urine, and increased frequency of urination. Upon diagnosis of bladder cancer, approximately 70% to 75% of patients exhibit non-muscle-invasive bladder cancer (NMIBC), while the remaining 25% to 30% manifest either muscle-invasive bladder cancer (MIBC) or distant metastases [2,3].

Bladder cancer intricately forms a highly immune-suppressive microenvironment, evading immune responses through diverse mechanisms. The bladder tumor microenvironment is characterized by a highly immune-suppressive milieu, including the accumulation of various types of immune cells with immunosuppressive phenotypes, such as regulatory T cells (Tregs), tolerogenic dendritic cells (tDCs), and tumor-associated macrophages (TAMs), along with an increased expression of programmed death-ligand 1 (PD-L1) and abnormal metabolism [4,5,6].

Therapeutic options for bladder cancer include chemotherapy and radical cystectomy for clinically localized cases, while systemic chemotherapy is available for patients with metastatic disease. Despite these aggressive treatment approaches, many patients do not respond to treatment or they experience recurrences, resulting in an unfavorable prognosis [7,8]. Due to immune suppression and tolerance caused by tumors, the clinical efficacy of cancer immunotherapy has been limited. Therefore, a deeper comprehension of immune evasion mechanisms in bladder cancer is necessary as it will play a crucial role in inhibiting recurrence and accurately identifying potential therapeutic targets.

Over the last 20 years, various groundbreaking immunotherapy strategies have been spotlighted in cancer treatment, such as PD-L1/PD-1 checkpoint inhibitors [9,10]. Nonetheless, there remains an unresolved challenge: most bladder cancer patients do not respond to immunotherapy across all stages of the disease. Therefore, there is a need to review candidate factors for overcoming resistance to immunotherapy in bladder cancer. This review aims to clarify how bladder cancer evades the immune system and investigates pathways and molecular targets that could overcome resistance to immunotherapy.

## 2. Bladder Cancer Therapy: Conventional Therapy and Immunotherapy

### 2.1. Standard Therapy of Bladder Cancer

Treating NMIBC offers a significant advantage over treating MIBC. This advantage lies in the ability to administer non-systemic chemotherapy directly into the bladder’s lumen, known as intravesical chemotherapy, thereby safeguarding normal body tissues from the potential toxicity of the agents used. Moreover, when combined with a transurethral resection of the bladder, intravesical chemotherapy has been proven to be highly effective in minimizing disease recurrence [11,12]. The most frequently utilized agents for this purpose include mitomycin C, epirubicin, thiotepa, gemcitabine, and doxorubicin. While these substances demonstrate effectiveness, there is still potential for enhancing their efficacy [13]. To improve efficacy, researchers have aimed to increase the absorption of chemotherapy by cancer cells following intravesical application. For instance, the utilization of albumin-bound nanoparticles to transport rapamycin or paclitaxel has been explored in bladder cancer treatment [14,15]. Preliminary animal studies have demonstrated that administering paclitaxel encased in polymers can reduce tumor weight [16]. Currently, the established treatment for MIBC involves the administration of neoadjuvant chemotherapy (NAC) before radical cystectomy (RC) [17,18,19]. Even with RC and NAC, 50% of patients still experience the development of distant metastases [20,21]. Additionally, adverse reactions to chemotherapy can limit NAC use among most patients [21]. Therefore, new treatments tailored for bladder cancer patients need to be developed.

### 2.2. Immunotherapy of Bladder Cancer

Presently, intravesical Bacillus Calmette-Guérin (BCG) therapy remains the foremost adjuvant therapy for high-risk NMIBC [22]. BCG, originally a tuberculosis (TB) vaccine, comprises live attenuated *Mycobacterium tuberculosis*. During the initiation phase of an immune response, BCG functions as a pathogen-associated molecular pattern (PAMP), stimulating pattern recognition receptors (PRRs) present on diverse cells, including bladder tumor cells and antigen-presenting cells (APCs), such as macrophages, dendritic cells (DCs), and others [8,23].

Most visible NMIBC can be removed through an endoscopic surgery known as transurethral resection (TUR). To address the high recurrence rates of NMIBC, intravesical therapies such as BCG can be utilized after TUR [7,24]. While BCG has remained a primary immunotherapy for many years, a considerable proportion of patients with NMIBC eventually transition to a state in which they do not respond effectively to BCG treatment [25,26]. Patients who do not achieve a complete response after BCG induction experience reduced five-year survival rates. Moreover, additional BCG therapy has been found to be ineffective [11,27,28]. BCG is used worldwide because certain chemotherapy options, such as mitomycin-C, may not be available in all countries [29]. Up to 35% of bladder cancer patients are treated with intravesical BCG, but 40 to 60% will have a recurrence of the tumor within 2 years [30].

Recent studies have demonstrated that immune checkpoints may play a role in the mechanism of BCG resistance [31]. The role of immune checkpoints involves preserving self-tolerance and regulating the magnitude and duration of inflammatory responses. However, their normal function in regulating this immune homeostasis is induced in cancer cells to evade immune attacks. Multiple antibodies that can block immune checkpoints have been approved for treating melanoma, lung cancer, and renal carcinoma, targeting additional immune checkpoints clinically [32]. In the ongoing randomized clinical trial KEYNOTE-057, pembrolizumab, a PD-1 inhibitor, has shown promising results in treating high-risk BCG-unresponsive NMIBC, leading to its recent FDA approval [33,34,35]. In a cohort of 96 patients, Balar et al. reported that an intravenous injection of pembrolizumab alone resulted in complete remission (CR) in 39 cases, but among them, 51% exhibited recurrent disease [36]. And recently, Atezolizumab, a monoclonal antibody that binds to PD-L1, was approved for the treatment of patients with MIBC [37]. However, there is a limitation that only MIBC patients who are not suitable for chemotherapy are included and there is no official maintenance treatment yet [37]. Therefore, additional evidence is needed to establish the safety and efficacy of immunotherapy in the treatment of bladder cancer and new therapeutic strategies are needed.

## 3. Factors That Bladder Tumors Exploit to Avoid Immune Responses

### 3.1. Immune Checkpoint Molecules (PD-1/PD-L1/CTLA-4)

Enhancing immunotherapies for bladder cancer will require deeper insights into the tumor microenvironment (TME) and effective techniques for leveraging the immune system against tumor progression. Immune checkpoints play a role in maintaining self-tolerance and controlling the intensity and duration of inflammatory reactions. However, tumors can exploit these pathways to evade the body’s immune response. In the past decade, immune checkpoint therapy has significantly transformed the treatment approach for bladder cancer [38,39,40]. Immune checkpoint therapy can stimulate an anti-tumor response by restoring T cell function through complex interactions among immune checkpoint molecules, such as PD-1, its ligand (PD-L1), and cytotoxic T lymphocyte-associated protein (CTLA-4) [32].

Expressed in activated T lymphocytes, PD-1 can regulate T cell effector functions and serve as a brake on T cell immune responses. Binding PD-1 to its counterpart, PD-L1, can lead to T cell exhaustion, a process that can reduce T cell activity (Figure 1). This is considered a mechanism to prevent auto-immunity [41]. The discovery of the involvement of the PD-1/PD-L1 axis in regulating T cells has led to preclinical studies demonstrating the overexpression of PD-1/PD-L1 in malignant cells [42]. Subsequently, PD-1/PD-L1 expression has been found to be higher in tumor tissue samples of bladder cancer patients than in normal tissues [41,42,43,44]. Patients unresponsive to BCG treatment show higher PD-L1 expression in both tumor and immune cells than those who are responsive to BCG treatment [45]. RNA sequencing analyses have revealed higher baseline PD-L1 levels in patients unresponsive to BCG [31,45]. The overall PD-1/PD-L1 pathway appears to be associated with localized bladder cancer immune response, indicating its value as a potential therapeutic target.

Expressed on activated T cells, CTLA-4 is a surface molecule that can interact with the B7 ligand found on B lymphocytes, DCs, and macrophages. While its full mechanism remains unclear, this element can exert a negative influence on the immune response. One proposed explanation is its structural similarity to CD28, suggesting that it may compete with CD28 for binding to ligands [46,47,48]. Other studies have shown that CTLA-4 expression is particularly high in MIBC and correlated with tumor size [49]. Currently, in addition to the already approved pembrolizumab (anti-PD-1), clinical trials for BCG non-responsive patients are ongoing. Typically, durvalumab is a human monoclonal IgG kappa antibody against PD-L1 and does not cause antibody-dependent cytotoxicity [50]. Clinical trials are being conducted on its efficacy in NMIBC patients, and it has been shown to be safe in combination with BCG [51]. There is also an ongoing phase I/II RIDEAU study aimed at determining whether systemic durvalumab in combination with anti-CTLA-4 tremelimumab is effective in patients with NMIBC [52]. The maximum tolerated dose (MTD) of durvalumab and 1-year HG relapse-free rate are the primary endpoints. There are no exact success figures for the ongoing combination treatment clinical trials because they are ongoing.

### 3.2. Human Leukocyte Antigen G (HLA-G)

HLA-G belongs to a non-traditional group of HLA class I molecules. It possesses immune-regulating characteristics [53]. HLA-G was initially reported to play crucial roles in maternal–fetal tolerance and tissue transplantation [54]. Since then, it has emerged as a significant immune checkpoint in various cancers, including bladder cancer, stomach cancer, colon cancer, and breast cancer [55,56,57,58]. Its inhibitory effect surpasses those of other immune checkpoints, as it can directly interact with key receptors, immunoglobulin-like transcript 2 (ILT2) and ILT4, thereby interfering with multiple stages of antitumor responses [59]. ILT2 exhibits distinct expression patterns on natural killer (NK) cells, T cells, B cells, monocytes, and DCs, while ILT4 is specific to myeloid cells [60,61] (Figure 1). Consequently, HLA-G interactions have the potential to disrupt initial or subsequent stages of immune responses [60]. Notably, HLA-G-induced suppressor cells such as Tregs and tDCs can perform long-term immunoregulatory functions, potentially blocking immunity [61]. Additionally, previous studies have demonstrated that HLA-G can promote cancer metastasis by inducing angiogenesis [62,63]. The expression of HLA-G in bladder cancer strongly correlates with tumor aggressiveness and poor survival outcomes [55].

### 3.3. Toll-like Receptor (TLR) Polyimorphism

TLRs are pattern recognition receptors (PRRs) that detect pathogen-associated molecular patterns (PAMPs) and initiate immune response. Upon TLR activation, NF-κB translocates to the nucleus and initiates the transcription of genes associated with cell proliferation, including cyclin D1, cyclin D2, c-Myc, and BCL-2 [64]. TLRs are expressed on APCs, such as macrophages and DCs, as well as on tumor cells [65,66]. The binding of ligands to TLRs induces APC maturation and the release of cytokines (TNF-α, IL-6, IL-12, and IFN) [65]. This is followed by the activation of NK cells and cytotoxic T cells, ultimately leading to the death of tumor cells [65]. However, TLR expression on bladder cancer cells can contribute to tumor growth and immune evasion by upregulating the growth factors and anti-apoptotic protein production [65,67]. Effective options achieved through TLR modulation may include blocking TLR involvement in inflammatory diseases and recruiting TLR signaling pathways against cancer.

The association between TLR polymorphisms and genetic susceptibility to various types of cancer has been extensively studied [68]. Hundreds of single-nucleotide polymorphisms (SNPs) have been identified in TLRs, but their functional consequences remain largely unknown. Nevertheless, several correlations between TLR polymorphisms and cancer have been reported [68,69]. Understanding TLR polymorphisms and their role in cancer will be helpful in selecting appropriate treatments. For example, the prevalence of TLR4 +3725GC and CC genotypes was found to be significantly increased in bladder cancer cases compared to controls [70]. TLR polymorphisms could potentially serve as therapeutic targets and prognostic indicators in bladder cancer.

### 3.4. Lactate

Cancer cells rely on aerobic glycolysis, known as the Warburg effect, to sustain growth even under normal oxygen conditions [71]. Several studies have demonstrated that increased lactate levels within the TME due to metabolic changes create an immunosuppressive environment [72,73]. Lactate dehydrogenase-A (LDH-A) plays a crucial role as an enzyme responsible for the final step of glycolysis, converting pyruvate into lactate. Elevated levels of LDH-A have been observed in various cancer types, such as cervical cancer and renal cell carcinoma [74], showing a strong association with cancer progression, metastasis, recurrence, and unfavorable clinical outcomes [74,75]. Lactate produced metabolically by LDH-A is considered an oncometabolite. Increased lactate levels in cancer are linked to a higher incidence of metastases and decreased survival [76]. Recent studies suggest that lactate contributes to the creation of an immunosuppressive TME by reducing the presence of NK cells and cytotoxic CD8^+^ T cells while suppressing the expression of interferon-γ (IFN-γ) [73] (Figure 1). Additionally, it has been reported that LDH-A-derived lactate within the TME can upregulate the expression of PD-L1 in lung cancer [77]. In bladder cancer, the overexpression of LDH-A has been correlated with tumor development and an unfavorable prognosis of patients [78]. Therefore, regulating LDH-A/lactate may be an effective approach to enhance the efficacy of anti-PD-1 therapy.

### 3.5. Prostaglandin E2 (PGE2)

PGE2, a bioactive lipid mediator, significantly contributes to immune evasion within the TME [79,80]. It interferes with anti-tumor immune responses by affecting various immune cells, including T cells [81], DCs [82,83], macrophages [84,85], and NK cells [86,87] (Figure 1). Regarding T cells, PGE2 can suppress the activation and proliferation of cytotoxic T cells and foster an immunosuppressive environment by inducing the expansion of Tregs [88,89,90]. Additionally, PGE2 can promote immune tolerance and tumorigenesis by inducing the polarization of macrophages towards the M2 phenotype [91]. PGE2 can also exert molecular effects by inhibiting effector T cell function and upregulating immune checkpoint molecules such as PD-L1 [92]. Increased levels of cyclooxygenase-2 (COX2), an essential enzyme in the biosynthetic pathway of PGE2, have been reported in various cancer types, including bladder cancer [93]. Recent studies suggest that COX2/PGE2 signaling plays a crucial role in the proliferation and regeneration of bladder cancer cells [94]. Considering these findings, appropriately regulating the metabolic process of PGE2 in bladder cancer could prove beneficial for immune-checkpoint-mediated treatment.

### 3.6. Tolerogenic Dendritic Cells (tDCs)

The TME encompasses various cell types collaborating in complex mechanisms, allowing cancer cells to evade immune surveillance. In addition to inhibition mediated by molecules such as PD-1 and CTLA-4, myeloid lineage cells can form a network of immunosuppressive cells, hindering the development of antitumor immunity [6]. This network includes TAMs and DCs. DCs primarily function to capture and present antigens, initiating the activation of CD4^+^ and CD8^+^ T cells. For the effective activation of T cells, DCs undergo a maturation process that involves the increased expression of antigen-presenting molecules (MHC class Ⅰ and Ⅱ), the upregulation of costimulatory molecules such as CD80 and CD86, and the production of immune-stimulating cytokine [95]. However, tumor cells typically employ various immunosuppressive strategies in their microenvironment to interfere with this process. Bladder cancer can influence the differentiation/maturation process of DCs, preventing the upregulation of co-stimulatory molecules and converting them into tDCs [96]. Recent studies have demonstrated that DCs exposed to Sialyl Tn (STn)-expressing bladder cancer cells show a decreased capacity to stimulate and guide T cells toward Th1 phenotypes, compromising their ability to generate an effective anti-tumor response [96,97,98,99].

### 3.7. Tumor-Associated Macrophages (TAMs)

TAMs exhibit diverse functions within the TME. The M1 polarization of TAMs can produce pro-inflammatory cytokines known to activate cytotoxic T cells and induce tumor immune responses. Conversely, M2 polarization can selectively attract regulatory T cells (Tregs) and Th2 cells while suppressing T cell responses [100]. IL-10-positive TAMs display an immunosuppressive phenotype similar to M2 macrophages in the context of MIBC, infiltrating the TME and suppressing immune responses [101] (Figure 1). The release of IL-10 by TAMs additionally heightens the expression levels of various immune checkpoints and co-inhibitory receptors (such as PD-L1, CTLA-4, and TIM-3). Compositions of chemokines within the TME substantially shape the direction and characteristics of macrophages. Specifically, chemokine (C-C motif) ligand 2 (CCL2) plays a crucial role in recruiting TAMs and contributes to bladder cancer cell proliferation, metastasis, and the establishment of an immunosuppressive environment in the TME [102,103]. Infiltrating TAMs can promote tumor development by altering the microenvironment of bladder cancer through the action of C-X-C motif chemokine ligand 8 (CXCL8) [103]. Conversely, CXCL8 has been shown to trigger pro-angiogenic and anti-apoptotic pathways when it is secreted by tumor cells [104,105,106].

### 3.8. Regulatory T Cells (Tregs)

Tregs play a crucial role in aiding tumors to evade anti-tumor immune responses and circumvent therapeutic interventions across various types of cancers [107,108]. A previous study has shown a marked increase in Tregs levels in peripheral blood samples of patients with bladder cancer [97]. In the same vein, the infiltration of T cell subpopulations within the tumor tissue showed significant reductions in tissue CD3^+^, CD4^+^, and CD8^+^ cells [109,110]. Patients with MIBC expressing forkhead box P3 (Foxp3), a transcription factor for Tregs, have a lower survival rate than patients with Foxp3-negative cancers [110]. Quinn et al. (2008) analyzed the effect of Treg inactivation before BCG vaccination on the development of protective immunity in a mouse model [111]. Although Treg inactivation increased the number of IL-2- or IFN-γ-producing CD4^+^ lymphocytes, the presence of CD4^+^CD25^+^ Tregs did not significantly affect the protective efficacy of the BCG vaccine [111]. However, the simultaneous suppression of Th2 cells and Tregs has been shown to increase Th1 responses, enhancing host protective immunity and BCG-induced vaccine efficacy [112]. Although our understanding of the impact of Tregs on bladder cancer remains incomplete, current efforts focus on tumor-infiltrating Tregs as a strategy for cancer therapy.

### 3.9. Cancer-Associated Fibroblasts (CAFs)

CAFs can produce key growth factors and cytokines as components of the ECM to promote tumor growth, metastasis, chemoresistance, and immune responses. Previous reports have revealed several mechanisms linking CAFs to immune evasion in cancer [113,114,115]. CAFs can remodel the extracellular matrix and secrete various factors such as cytokines, chemokines, and growth factors. These factors can influence the recruitment, function, and polarization of immune cells [116,117,118,119] and create an immunosuppressive environment that promotes tumor progression [120,121,122,123]. CAFs have been found to modulate immune checkpoint molecules such as PD-L1 [124,125]. The interaction between PD-L1 on CAFs and its receptor PD-1 on immune cells, such as T lymphocytes, can lead to immune suppression by inhibiting the activity of these immune cells, allowing tumors to evade immune attacks [126,127]. One study showed that a subset of CAFs, namely iCAFs, play an important role in promoting tumorigenesis or the migration of primary tumors by promoting angiogenesis and the epithelial–mesenchymal transition (EMT) [128]. Additionally, studies have confirmed that IL-6 is upregulated in bladder cancer iCAFs, indicating that the expression of IL-6R in bladder cancer cells is suitable for responding to the IL-6 cytokine secreted by iCAFs [129]. Siltuximab, an IL-6 blocking antibody, has already been approved by the FDA and has demonstrated antitumor efficacy against ovarian, prostate, and lung cancer [130]. Clinical trials investigating targeted approaches, such as IL-6 signaling inhibition in bladder cancer, are still lacking. However, this approach to disrupt CAF function may be a clinically promising strategy. C-X-C motif chemokine 12 (CXCL12) derived from CAFs has been established as one of the most potent chemokines that can regulate immune cells [131]. Understanding the intricate interplay between bladder cancer cells, CAFs, and the immune system is crucial for developing more effective therapeutic strategies.

## 4. Potential Therapeutic Strategies for Modulating Immune Evasion

### 4.1. Therapeutic Combination with Immune Checkpoint Inhibitor (ICI)

Currently, in the field of treating NMIBC, the main supporting therapy is intravesical BCG, but patients unresponsive to BCG undergo ICI treatment [132]. For patients diagnosed with MIBC, common treatment options include cisplatin chemotherapy or radical cystectomy. Nevertheless, there are instances in which patients are unsuitable for chemotherapy or choose not to undergo cystectomy [133]. In such cases, ICIs can serve as an alternative treatment method for MIBC patients [133]. Especially in the context of MIBC, ICI stands out as a significant treatment option due to its lower rates of recurrence and survival rates, even after cystectomy [134]. As of 2023, the FDA has approved three drugs (pembrolizumab, valrubicin, and nadofaragene firadenovec-vcng) for patients unresponsive to BCG and either ineligible for chemotherapy or refusing radical cystectomy [133].

Despite increased PD-L1 expression in bladder cancer patients, a significant number of patients do not respond to PD-1/PD-L1 inhibitors [135,136,137]. Therefore, the need for more accurate biomarkers is emphasized.

Combination therapy can be a strategy to enhance the efficacy of PD-1/PD-L1 inhibition (Figure 2). To date, the FDA has approved pembrolizumab only for patients with NMIBC and BCG failure who are unwilling or unable to undergo RC [138]. There is no approved combination administration method with ICI yet, but combination administration with BCG or ICI has shown effectiveness in clinical trials. Atezolizumab is a monoclonal antibody that inhibits the PD-L1/PD-1 pathway. In NCT02792192, 33.3% of the atezolizumab-only group and 41.7% of the atezolizumab + BCG group achieved complete remission (CR) [139]. The combination of CTLA-4 and PD-L1 inhibitors can result in heightened levels of CD4^+^ and CD8^+^ T cells, decreased Tregs, the elevated production of pro-inflammatory cytokines, and a shift in macrophage polarization from M2 to M1 [140,141]. The most recent NABUCCO clinical trial observed that in patients with stage Ⅲ bladder cancer who were not suitable for cisplatin therapy, the administration of preoperative ipilimumab (anti-CTLA-4) and nivolumab (anti-PD-1) before RC resulted in a pathological complete response (pCR) rate of 46% [125,142].

### 4.2. Therapeutic Combination with HLA-G Target

In most cases, the HLA-G/ILT interaction can promote cancer cells to evade immune surveillance and anti-tumor immunity [143,144]. Therefore, a concurrent blockade of PD-1/PD-L1 and HLA-G/ILT2 might offer potential benefits in specific patient populations (Figure 2). Several in vitro studies have demonstrated that NK cells can more effectively eliminate HLA-G-negative leukemia, glioma, ovarian carcinoma, and hepatocellular carcinoma cell lines than those transfected with HLA-G [145,146,147,148]. HLA-G is frequently but heterogeneously expressed in various cancers, including bladder cancer. This expression pattern varies depending on tumor subtypes, tumor grades, tumor stages, and compositions of immune cell infiltration [149]. Therefore, further research is needed to explore the specific immunological functions of HLA-G and its potential therapeutic targeting capabilities. However, HLA-G has been proven to upregulate the expression of PD-1 on T lymphocytes. Thus, blocking HLA-G/ILT2-related PD-1/PD-L1 interactions might be an effective treatment strategy for bladder cancer [150].

### 4.3. TLR-Targeted Modulation

The expression and polymorphism of TLRs in bladder cancer cells are closely related to tumor development [65,70,151]. The activation of TLRs on tumor cells can trigger both pro- and anti- tumor responses, and the use of TLR antagonists and agonists has been discussed extensively [152,153,154,155]. Studies to prevent the immune evasion of cancer cells through TLR antagonists have been conducted in various carcinomas [156]. Indeed, significant tumor-destroying properties have been demonstrated for breast and ovarian carcinomas. [156]. However, some studies have reported more direct cytotoxic effects and in vivo anti-tumor effects of TLR agonists on bladder cancer cell lines compared to TLR antagonists [151,157].

Recent clinical studies on TLR agonists in cancer have shown promising results, especially when combined with other immunotherapies [158]. For instance, the combination of recombinant cancer vaccines with TLR3, 4, and 9 agonists has exhibited the ability to boost immune stimulation and enhance T cell responses in melanoma patients [159,160].

BCG, acting as a TLR2 and TLR4 agonist, remains a cornerstone in immunotherapy for NMIBC patients. However, alternative options are needed for patients unresponsive to BCG. Recently, imiquimod and TMX-202 (TLR7 agonists) were tested for bladder cancer immunotherapy, while CpG ODN (a TLR9 agonist) and HP-NAP (a TLR2 agonist) demonstrated reduced tumor growth in an MB49 bladder cancer mouse model [161,162,163,164]. As discussed earlier, genetic susceptibility to bladder cancer may vary depending on the TLR polymorphism. Studies indicate that variant alleles of TLR2 (ID+DD) are associated with disease susceptibility and risk of progression in bladder cancer [70]. Furthermore, the prevalence of TLR4 +3725G/C has been significantly increased in bladder cancer cases [165]. Therefore, these TLR polymorphisms could potentially serve as new therapeutic targets and prognostic indicators in bladder cancer.

### 4.4. LDH-A/lactate

Metabolites present within the TME significantly interfere with the function of immune cells, especially anti-tumor T cells. Therefore, therapeutic approaches targeting metabolic changes to improve the TME may provide promising complementary treatments in combination with ICI. According to a recent study, increased LDH-A in melanoma patients is correlated with a decreased response to anti-PD-1 immunotherapy [76]. In a mouse model of melanoma, blocking LDH-A can increase the production of IFN-γ and granzyme B in NK cells and CD8^+^ T cells [166]. Additionally, it can enhance anti-tumor immune response to immune checkpoint inhibitors [166]. Tumors with LDH-A deficiency exhibit notably higher levels of responsiveness to anti-PD-1 treatment than tumors expressing LDH-A [76]. This combination therapy can potently block the PD-1/PD-L1 pathway, thereby enhancing the infiltration and activation of NK cells and CD8^+^ T cells and increasing anti-tumor inflammatory responses (Figure 2).

LDH-A is upregulated in MIBC, promoting the proliferation of MIBC cells [78]. Therefore, it can be a viable option not only for treating NMIBC through combination treatment, but also for addressing MIBC, which generally has a less favorable prognosis.

### 4.5. mPGES1/15-PGDH

PGE2 contributes to tumor progression through a variety of mechanisms, including immunosuppression within the tumor environment and the proliferation or regeneration of cancer stem cells [92,167]. Elevated levels of COX2 have been documented in various cancers, including bladder cancer [93,168]. Recent studies have shown that inhibiting COX2 can reduce tumor-induced Treg activity and ultimately restore anti-tumor responses [169]. However, persistent COX2 inhibition can lead to adverse cardiovascular and gastrointestinal effects [170,171]. Therefore, targeting PGE2-producing enzyme microsomal prostaglandin E synthase-1 (mPGES1) and employing a targeted genetic enhancement of NAD^+^-dependent 15-hydroxyprostagladin dehydrogenase (15-PGDH), an enzyme that decomposes PGE2, might offer more efficient and safer approaches to combating cancer [92] (Figure 2). mPGES1 stands as the pivotal terminal enzyme implicated in the production of PGE2. Newer research has indicated that suppressing mPGES1 activity can lead to decreased PD-L1 expression in myeloid cells that infiltrate bladder tumor tissues [92]. Therefore, focusing on PGE2 metabolism may help reduce PD-L1-mediated immunosuppression.

### 4.6. Sialyl Tn (STn)

STn, a widespread tumor-associated carbohydrate antigen, is detected in more than 80% of human carcinomas but rarely found in normal tissues [172]. Within bladder cancer, STn is associated with the malignancy of the disease by amplifying the mobility and invasive potential of cancer cells [99]. STn can interfere with the expression of MHC-class Ⅱ and co-stimulatory molecules within DCs, reducing their ability to display cancer-related antigens to T cells [96] (Figure 1). Consequently, DCs become unresponsive to subsequent activation signals. Additionally, it can establish a tolerogenic microenvironment that enables cancer to evade attacks from both innate and adaptive immunity by impairing effective anti-tumor responses of Th1 cells [96,99]. DCs are used in current anti-tumor cell vaccines. The common strategy is injecting tumor cell antigens into a patient’s DCs [173]. Hence, merging therapies that can induce Th1 responses with DC-based treatments targeting cancer cells bearing STn holds promise as a strategy to stimulate protective anti-tumor responses [96] (Figure 2).

A study has revealed higher STn expression in metastatic MIBC compared to normal tissue, indicating a robust connection between STn antigens and tumor metastasis. This insight offers crucial molecular information for targeting aggressive bladder cancer cells [174]. Although additional research involving a larger cohort of patients is necessary before translating this into clinical practice, it holds potential as a valuable target for treating MIBC. Given its poor prognosis and limited therapeutic options compared to NMIBC, further exploration is warranted.

### 4.7. Regulation of Immune Cells within the TME through Chemokine Control

Chemokines and their receptors are recognized as mediators of chronic inflammation. They are important factors contributing to the progression of cancer [175]. Chemokines are increasingly recognized for their involvement in attracting a variety of cell types into the complex environment of tumors and inducing the spread of cancer cells to new sites. These encompass infiltrating cells such as TAMs and CAFs [175,176,177]. Therefore, blocking chemokines in the TME could be a promising approach to enhancing the efficacy of immunotherapy. CXCL12, also known as stromal cell-derived factor 1 (SDF1), is a chemokine crucially expressed across various tissues to maintain homeostasis. Its involvement in the development of bladder cancer has been extensively documented and discussed [178]. Within the TME, interaction between CXCL12 and its primary receptor C-X-C chemokine receptor type 4 (CXCR4) triggers downstream signaling pathways, leading to various outcomes, including the proliferation and evasion of immune cells. In a CXCL12-dependent manner, CAFs can effectively reduce the migration of CD8^+^ T cells towards the stromal region adjacent to the tumor, consequently limiting the infiltration of CD8^+^ T cells around tumor cells [179]. CXCL12 has the ability to rapidly increase the expression of PD-L1 in cancer cells within a short timeframe [131]. Suppressing CXCL12 produced by CAFs can notably reduce the impact of CAFs on the expression of PD-L1 [131] (Figure 2). A comprehensive screening of CAFs will offer deeper insights into the development of bladder cancer. Directing therapeutic efforts towards CXCL12 derived from CAFs could potentially be a valuable strategy for treating bladder cancer.

CCL2, a significant chemokine crucial for macrophages, exhibits high expression within cancer cells. It is actively secreted by these cells [102]. Several research studies have shown that CCL2 can promote the M2 polarization of TAMs, consequently amplifying immunosuppressive properties within the TME [180,181]. It has been found that CCL2 overexpression in bladder cancer is due to the FAM171B protein, which can act as an immune regulator and upregulate CCL2 expression [102]. Altering the immune microenvironment within tumors by targeting FAM171B could potentially serve as a viable strategy to increase the efficacy of immunotherapy when managing bladder cancer (Figure 2). Additionally, potential therapeutic approaches for bladder cancer involve the targeting of the CCL2-CCR2 axis, which could have encouraging results (Figure 2).

The CXCL8/CXCR2 pathway is crucial in human cancer. It contributes to tumor progression through various mechanisms [182]. Bladder cancer patients show notable increases in TAM infiltration than patients in a normal control group. Moreover, TAM levels show positive correlations with the expression of CXCL8 in bladder cancer tissues [104,183]. CXCL8 produced by TAMs has the potential to induce the expression of VEGF in bladder cancer cells. This stimulation contributes to changes in bladder cancer cell migration and invasion, consequently promoting bladder cancer progression [183]. Moreover, the engagement between bladder cancer cells and endothelial cells (ECs) can augment EC recruitment via the CXCL8–CXCR2 pathway [184]. Vascular ECs can interact with bladder cancer cells and tissues to promote cancer progression by activating the epidermal growth factor receptor (EGFR) signaling pathway [184]. This pathway can induce cancer proliferation, migration, and the secretion of CXC chemokines from bladder cancer cells. Therefore, targeting CXCL8 signaling presents a potential and promising therapeutic strategy for managing bladder cancer.

## 5. Conclusions

Immunotherapy for cancer has garnered significant attention as an exceedingly effective and promising approach. Despite notable progress in the development, clinical trials, and validation of therapeutics such as immune checkpoint inhibitors, gene-manipulated immune cells, and novel cancer vaccines, the clinical efficacy of cancer immunotherapy remains limited. The TME consists of a variety of cells that collaborate to assist cancer cells in evading immune surveillance and surviving therapeutic interventions. Within the TME, receptor–ligand interactions, metabolic byproducts, and the genetic sensitivity of these molecules can closely interact. Therefore, concurrently addressing factors involved in tumor-induced immune suppression during immunotherapy administration has the potential to further amplify anti-tumor immune responses. New insights into immune evasion in bladder cancer, as discussed earlier, suggest the potential to enhance cancer immunotherapy by targeting specific immune suppressive elements within the TME.

## Figures and Tables

**Figure 1 ijms-25-03105-f001:**
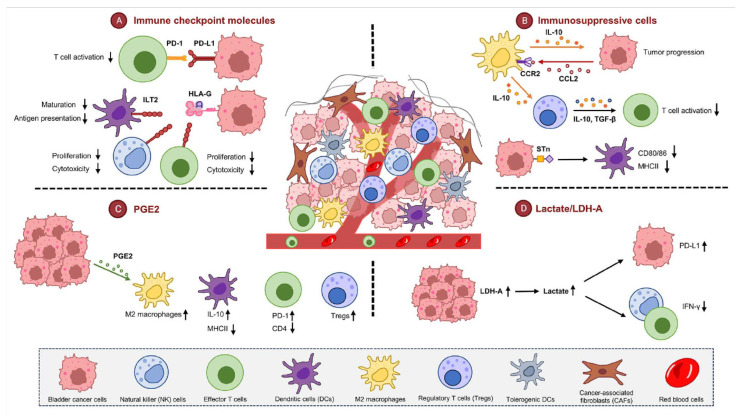
Bladder cancer evades immune responses by forming an immunosuppressive microenvironment. (A). PD-1 regulates T cell effector functions and acts as a brake on T cell immune responses. HLA-G-induced inhibitory cells have long-term immune regulatory functions, inducing the generation of suppressive cells and blocking immune effector mechanisms. (B). IL-10-positive TAMs showed an immunosuppressive phenotype resembling that of M2 macrophages within the context of MIBC. STn interferes with the expression of MHC-Ⅱ and co-stimulatory molecules in DCs, reducing their ability to display cancer-associated antigens to T cells. (C). PGE2 interferes with anti-tumor immune responses by affecting the function of immune cells such as T cells, DCs, macrophages, and NK cells. (D). Lactate contributes to creating an immunosuppressive TME by reducing the presence of NK cells and cytotoxic T cells, while suppressing the expression of IFN-γ. PD-1, programmed cell death protein 1; HLA-G, human leukocyte antigen G; TAMs, tumor-associated macrophages; MIBC, muscle-invasive bladder cancer; STn, Sialyl Tn; PGE2, prostaglandin E2; DCs, dendritic cells; NK cells, natural killer cells; IFN-γ, interferon- γ.

**Figure 2 ijms-25-03105-f002:**
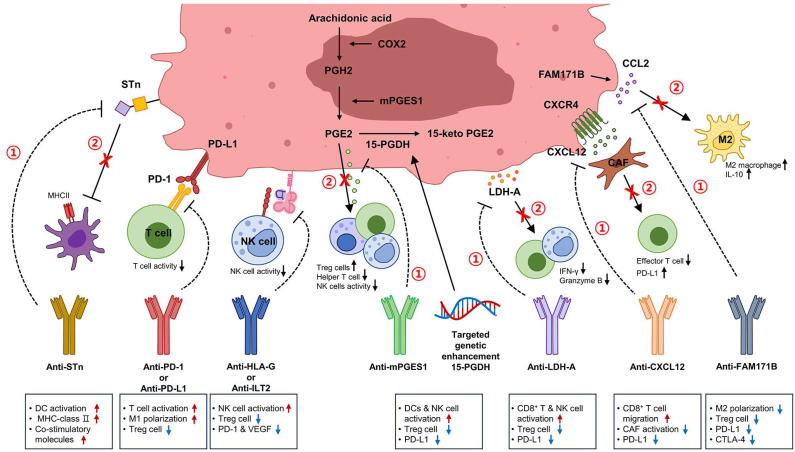
**Potential therapeutic strategies for modulating immune evasion.** This diagram summarizes candidate factors for treatment using the various mechanisms that bladder cancer uses to evade immune responses. (Red numbers indicate the sequence of results when the target is blocked, and an x means that the path is suppressed.) Targeting the STn can enhance the expression of MHC class II and costimulatory molecules in DCs and their ability to present antigens to T cells. PD-L1/PD-1 inhibitors can increase the levels of CD4^+^ and CD8^+^ T cells, reduce Tregs, and induce macrophage M1 polarization. Targeting HLA-G can upregulate PD-1 expression on T lymphocytes, block angiogenesis, and activate NK cells. Targeting mPGES1 and employing targeted genetic enhancement of 15-PGDH can reduce PD-L1-mediated immunosuppression. Blocking LDH-A can increase the production of IFN-γ and granzyme B in NK cells and CD8^+^ T cells and enhance anti-tumor immune responses to immune checkpoint inhibitors. Inhibiting CXCL12 produced by CAFs can significantly reduce the effect of CAFs on PD-L1 expression and promote the migration of CD8^+^ T cells toward the tumor stromal area. Targeting the FAM171B protein, which can upregulate CCL2 expression in bladder cancer, can suppress M2 polarization of macrophages and Tregs. STn, Sialyl Tn; DCs, dendritic cells; PD-L1, programmed cell death-ligand 1; PD-1, programmed cell death protein 1; Tregs, regulatory T cells; HLA-G, human leukocyte antigen G; NK cells, natural killer cells; mPGES1, PGE2-producing enzyme microsomal prostaglandin E synthase-1; 15-PGDH, PGE2-degrading enzyme NAD+ dependent 15-hydroxyprostaglandin dehydrogenase; LDH-A, lactate dehydrogenase-A; IFN-γ, interferon-γ; CXCL12, C-X-C motif chemokine 12; CAFs, cancer-associated fibroblasts; CCL2, chemokine(C-C motif) ligand 2; Tregs, regulatory T cells.

## Data Availability

No new data were created or analyzed in this study. Data sharing is not applicable to this article.

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
