# Peer review of "Strategies for Overcoming Immune Evasion in Bladder Cancer"

_ijms, 2024, doi:10.3390/ijms25063105_

Round 1
Reviewer 1 Report
Comments and Suggestions for Authors
I would like to congratulate the authors for their extensive work on this detailed review of immune evasion in bladder cancer.
I have just two minor comments:
1. I do not remember now by heart whether it was used or not, but I recommend the authors write the gene names in italics, which helps us to differentiate them from proteins, which are written with normal punto.
2. When I read the title of the review, I was expecting to find more detailed information on the drugs that aim at the reviewed immune evasion pathways. I wonder if they have given all the drugs/medications (either commercially available or still in the experimental phase) aiming at these pathways or not. If all drugs have not been included, it would be much better to enlarge the review by adding them. If they have been fully added, then as not too much information occupies in the review, it might be better to change the name of the review (by giving more emphasis that the pathways have been studied in this review).
Comments on the Quality of English LanguageThere are some minor grammar, spelling, and punctuation errors to be corrected.
Author Response
I have provided point-by-point responses to the comments from the reviewer below.
Comments 1: Thank you for your manuscript submission. The topic is very interesting. The manuscript is well-presented; However; I believe that the authors have completely ignored the role of toll-like receptors and the related SNPs in the field. As the manuscript is a Review, it is recommended to add a subtitle in association with TLRs and SNPs and their importance in the field. This point can enrich your manuscript as an effective review. In this regard, please do read and the following papers to References section of the manuscript to have a fruitful review:
- The role of Toll-like receptor (TLR) polymorphisms in urinary bladder cancer. Genetic Polymorphism and cancer susceptibility. 2021:281-317.
- Toll-like receptor-guided therapeutic intervention of human cancers: molecular and immunological perspectives. Front Immunol. 2023 Sep 26;14:1244345. doi: 10.3389/fimmu.2023.1244345. PMID: 37822929; PMCID: PMC10562563, and Toll-Like Receptors (TLRs) in Health and Disease: An Overview
In accordance with the aforementioned comments, it is necessary to revise the Conclusion section, too.
Response 1: I deeply appreciate your comments, and I resonate with your perspective. I believe that incorporating the involvement of Toll-like receptors in immune evasion mechanisms would enrich the paper. I have consulted the paper you provided and other relevant literature to investigate the link between Toll-like receptors (TLRs) and immune evasion in bladder cancer. It is noteworthy that Toll-like receptors (TLRs) in bladder cancer display dual properties by promoting both pro- and anti-tumor responses. In particular, TLRs expressed in bladder cancer have been reported to induce tumor growth and immune evasion. We noted that effective options achieved through TLR modulation may include blocking TLR involvement in inflammatory diseases and recruiting TLR signaling pathways against cancer. (page 4; lines 172-184)
We include information on the mechanisms of TLRs as well as treatment strategies. The ambivalence of TLRs has been described separately between TLR antagonists and agonists. In particular, we demonstrated that activation of TLRs complements enhanced antitumor immunity in bladder cancer, providing reference data. Recent clinical studies of TLR agonists in cancer have shown favorable results when combining these molecules with other immunotherapies. Therefore, the use of TLR agonists may also have therapeutic potential in bladder cancer (page 8; lines 362-375).
Based on the suggestions provided, we consulted information on Toll-like receptors (TLRs) and single nucleotide polymorphisms (SNPs). Upon confirmation, case reports and literature linking TLR polymorphisms and immune evasion have been extremely rare. So there were limitations in continuing the explanation. However, evidence for a correlation between TLR polymorphisms and prevalence has been reported and can be used as an important indicator of disease. TLRs are molecules that initiate and mature innate immunity, and are expressed not only in immune cells but also in cancer cells. Therefore, it is assumed that there may be a relationship between TLR polymorphisms and changes in immunity. Representatively, the prevalence of TLR4 +3725GC and CC genotypes was found to be significantly increased in bladder cancer cases compared to controls. (page 4; lines 185-193)
As a result, we attempted to suggest a direction for selective treatment based on polymorphisms and a TLR mechanism that can target this. BCG, used in patients with NMIBC, is a TLR2 and TLR4 agonist. However, alternative treatment options are needed for patients who do not respond. So TLR polymorphisms may be good candidates for new therapeutic targets and prognostic indicators in bladder cancer. We have added references and therapeutic potential addressing TLR polymorphisms in various carcinomas, including bladder cancer. (page 8; lines 376-386)
Reviewer 2 Report
Comments and Suggestions for Authors
This study was reported the strategies for overcoming immune evasion in bladder cancer. Overall, this paper is well written. The reviewer thinks that this report has useful information for readers. The reviewer would like to suggest some critiques as follows.
1. The prognosis and treatment strategies for NMIBC and MIBC are quite different. Therefore, I believe that they should be listed separately.
2. As for the molecules, they seem to be mainly associated with the ICI treatment of MIBC, so they should be emphasized. As for the text, I think it is fine, and it would be easier for the reader to understand if a few sentences were added at the beginning.
Author Response
I have provided point-by-point responses to the comments from the reviewer below.
Comments 1: This study was reported the strategies for overcoming immune evasion in bladder cancer. Overall, this paper is well written. The reviewer thinks that this report has useful information for readers. The reviewer would like to suggest some critiques as follows.
- The prognosis and treatment strategies for NMIBC and MIBC are quite different. Therefore, I believe that they should be listed separately.
Response 1: I appreciate your thoughtful comment, and I resonate with your perspective. Strategically describing the diagnosis and treatment of NMIBC and MIBC separately appears to be a valid approach, which will contribute to a more specific research paper. I have thoroughly researched the references and additional data used. In the manuscript, factors related to immune evasion were divided into sections corresponding to NMIBC and MIBC with regarding treatment strategies. It has been reported that LDH-A is upregulated in MIBC, promoting proliferation of tumor cells. Therefore, it can be a good choice not only for treating NMIBC through combination therapy but also for treating MIBC, which has a relatively poor prognosis. (page 9; lines 400-402) Additionally, one study found that STn expression was higher in metastatic MIBC compared to normal tissue, suggesting it may be a viable target for MIBC treatment, given its poor prognosis and limited treatment options compared to NMIBC. (page 10; lines 432-438)
However, there is an overall lack of research in the field of bladder cancer, especially in the area of immune evasion targets. Specifically, the pre-clinical research phase is primarily focused on immune checkpoint inhibitors (ICIs). Additionally, the current study provides a comprehensive description of bladder cancer without clearly distinguishing between NMIBC and MIBC. Because of this, there were limitations in specifically dividing treatment strategies. In our paper, we aimed to utilize the available comprehensive data on bladder cancer to contribute to treatment strategies. If research continues to advance based on the therapeutic directions of this review, I think we will be able to obtain more targeted treatment options, as you mentioned.
Comments 2: As for the molecules, they seem to be mainly associated with the ICI treatment of MIBC, so they should be emphasized. As for the text, I think it is fine, and it would be easier for the reader to understand if a few sentences were added at the beginning. Thank you for
Response 2: I appreciate your insights aimed at enhancing readers' understanding, and I resonate with your perspective. With your perspective in mind, we have compiled information about how immune checkpoint inhibitors (ICIs) are being utilized in both NMIBC and MIBC patients. Currently, the main adjuvant treatment for NMIBC is intravesical BCG, but ICI is administered in patients who do not respond to BCG. Patients with MIBC are typically offered cisplatin chemotherapy or radical cystectomy. However, some patients are not suitable for chemotherapy or refuse cystectomy. In such case, ICI can also be used as a treatment method for MIBC patients. In particular, we emphasized that ICI is an important treatment option for MIBC because recurrence and survival rates are limited even after cystectomy. (page 7; lines 319-329)
Reviewer 3 Report
Comments and Suggestions for Authors
Thi is a very nice, easy-to-read and informative review article, and it does apparently cover the latest developments in immuno-oncology.
The introduction provides some background information on the standard therapies currently used or bladder cancer, and with a 50% survival chance, there is much unmet need. The section on chemotherapy is rather small and somewhat superficial, it is obvious that the authors want to move quickly into the territory of immunotherapies.
nevertheless, the chemotherapy introduction could be expanded since this is most likely to be continued, or preceding, immunotherapies when these become more widely available for patients also in lower-income countries. It would also be nice to hear something about the number of immunotherapies currently used, such as BCG. How many patients do receive it at the moment? Is this already widely established in most countries?
Before entering section 3, which deals with the molecular immunology, it would be nice to offer the reader some numbers. How many patients are currently receiving which types of immunotherapies, what are the response rates, is there any information on overall survival? This may not be "molecular" information but still relevant, in my opinion, to illustrate the scope of the problem and the options at stake.
on a short note, HLA-G should get its own sub-chapter, just like PD1/PD-L1 should get, and also CTLA-4. These "classic" immune checkpoint inhibitors are covered with relatively little text. I could imagine that there is a lot of additional information out there that may be interesting for readers to browse. For example, are there any clinical trials ongoing or planed in bladder cancers, that will utilize monoclonal antibodies targeting PD-1, PD-L1, or CTLA4? [Edit: this is covered in chapter 4 but wouldnt it make sense to have the clinical studies in the same chapter as the description of the targets?]
And if, are they combined with other treatments, or with certain strategies for patient stratification before therapy? All of this is important since these therapies are already established, but still new enough to undergo several changes and fine-tuning measures in clinical application. None of this is covered. I can see that the authors may want to focus on newer developments, newer targets, but the "old" ones arent aged well yet and are still very interesting to the reader.
Then, generally, the topics that are covered are a bit unorganized. First, immune checkpoint inhibitors are covered, then HLA-G, then its shifting to signaling (PLG), next to metabolism (lactate), only to return to immune cell subpopulations (dendritic cells and TAMs). This is confusing and a bit chaotic, to say the least, and should be re-organized according to related topics. SOme of these "chapters" are also just a stub and are not informative, so they could either be expanded - or completely omitted, but they arent very satisfactory the way they are now. The best example is paragraph 3.6 on Tregs. Its only 7 lines, and doesnt cover the topic even remotely.
I do like, however, the chapter on CAFs, also since this is a very hot topic thats currently under intensive investigation. But even this is a bit on the short side-... the authors want to cover so many topics that in the end, nost of them are handled a bit superficially.
Chapter 4 then covers the clinical trials. I wonder why these are separated from the description of the targets. Wouldnt it be easier to read and to comprehend if, after the original (and short) description of the molecular target, the information on therapy and clinical trials would follow?
However, this chapter also brings in new players: the chemokines, which have not been mentioned previously. That also appears rather arbitrary: why are they mentioned here, and not already in chapter 3, like most of the other targets?
So, in conclusion, there is a lot of good content and its not difficult to read, but it couls still be significantly improved by focusing and making the entire story a more consistant narrative.
Author Response
I have provided point-by-point responses to the comments from the reviewer below.
Comments 1: This is a very nice, easy-to-read and informative review article, and it does apparently cover the latest developments in immuno-oncology.
- The introduction provides some background information on the standard therapies currently used or bladder cancer, and with a 50% survival chance, there is much unmet need. The section on chemotherapy is rather small and somewhat superficial, it is obvious that the authors want to move quickly into the territory of immunotherapies.
nevertheless, the chemotherapy introduction could be expanded since this is most likely to be continued, or preceding, immunotherapies when these become more widely available for patients also in lower-income countries. It would also be nice to hear something about the number of immunotherapies currently used, such as BCG. How many patients do receive it at the moment? Is this already widely established in most countries?
Response 1: I appreciate your thoughtful comment, and I resonate with your perspective. As you mentioned, our paper mainly deals with immune evasion in bladder cancer and is meaningful in providing insight into treatment strategies for it, so we focused on immunotherapy. I agree with your perspective. It would be a more detailed paper if it included the number of immunotherapies currently in use, including BCG, how many patients are receiving them, and whether they are widely established in most countries. First of all, BCG is used worldwide, including in the United States, Korea, Brazil, and Hong Kong, as a treatment for bladder cancer. In addition, we added the reference that 35% of bladder cancer patients use BCG, and that 40% to 60% of them relapse. (page 2; lines 92-95) Next, we added information about immunotherapy drugs used in addition to BCG. Currently, pembrolizumab is used for patients who do not respond to BCG, and atezolizumab was recently approved for use in patients with MIBC who are not suitable for chemotherapy. However, it was not clear exactly how many patients were using these two ICIs and what the survival rate was. Therefore, in the case of pembrolizumab, patient survival and recurrence rates were presented in clinical trials instead. (page 3; lines 103-111)
Comments 2: Before entering section 3, which deals with the molecular immunology, it would be nice to offer the reader some numbers. How many patients are currently receiving which types of immunotherapies, what are the response rates, is there any information on overall survival? This may not be "molecular" information but still relevant, in my opinion, to illustrate the scope of the problem and the options at stake.
Response 2: I agree with your perspective. Since HLA-G is also an immune checkpoint, it was grouped together with PD-1, PD-L1, and CTLA-4 and explained in separate paragraphs. However, as you said, it will be easier for readers to understand if we explain HLA-G separately from the traditional immune checkpoint. In the main text, HLA-G has been separated into its own subchapter. (page 4; line 156) We also added information about ongoing clinical trials for bladder cancer utilizing monoclonal antibodies targeting PD-1, PD-L1, or CTLA4. The addition of clinical trials for ICIs, in addition to the already approved anti-PD-1 drug pembrolizumab, may represent a current step in bladder cancer research. The reason why clinical trials and treatments for ICI are presented in Chapter 4 is to emphasize that ICI is more important in MIBC. However, I agree with the opinion that including clinical trials in the same chapter as the topic description may help readers' understanding. Therefore, after explaining the immune checkpoints, the clinical trials currently in progress were added and explained in more detail in Chapter 4. (page 3; lines 144-154)
Comments 3: And if, are they combined with other treatments, or with certain strategies for patient stratification before therapy? All of this is important since these therapies are already established, but still new enough to undergo several changes and fine-tuning measures in clinical application. None of this is covered. I can see that the authors may want to focus on newer developments, newer targets, but the "old" ones arent aged well yet and are still very interesting to the reader.
Response 3: Thank you for your helpful comments. Currently, pembrolizumab is administered alone to patients who do not respond to BCG treatment, and combined administration is still in clinical trials. So, I understand your opinion that even though combination therapy is still in clinical trials, it is new and important and should be added to the treatment strategy. As co-administration with BCG or other ICIs is attracting attention, clinical trials currently in progress have been added to the content. (page 8; lines 334-339)
Comments 4: Then, generally, the topics that are covered are a bit unorganized. First, immune checkpoint inhibitors are covered, then HLA-G, then its shifting to signaling (PLG), next to metabolism (lactate), only to return to immune cell subpopulations (dendritic cells and TAMs). This is confusing and a bit chaotic, to say the least, and should be re-organized according to related topics. SOme of these "chapters" are also just a stub and are not informative, so they could either be expanded - or completely omitted, but they arent very satisfactory the way they are now. The best example is paragraph 3.6 on Tregs. Its only 7 lines, and doesnt cover the topic even remotely.
Response 4: Thank you for your comment, and I agree with your point of view for a systematic paper. Title order has been modified to include immune checkpoint inhibitors (PD-1, PD-L1, CTLA-4), HLA-G, TLR (This is an immune evasion element added at the request of another reviewer, TLR expression on bladder cancer cells can contribute to tumor growth and immune evasion by upregulating the growth factors, and anti-apoptotic protein production), metabolic products (lactate), PGE2 (lipid mediator), and immune cell subgroups. Accordingly, the title classification of treatment strategies has also changed. Following your advice, I have strengthened the content about Tregs and added references. One study showed that Treg inactivation before BCG treatment increased the number of IL-2 or IFN-γ producing CD4+ lymphocytes in a mouse model of bladder cancer. In particular, simultaneous suppression of Th2 cells and Tregs has been shown to increase Th1 responses and improve BCG-induced vaccine efficacy. Although research on regulatory T cells in bladder cancer is lacking, we would like to suggest that this has several notable implications. (page 6; lines 265-275)
Comments 5: I do like, however, the chapter on CAFs, also since this is a very hot topic thats currently under intensive investigation. But even this is a bit on the short side-... the authors want to cover so many topics that in the end, nost of them are handled a bit superficially.
Response 5: I agree with you that CAF is currently being addressed intensively. One study showed that iCAFs, a subset of CAFs, play an important role in promoting tumor formation and migration by promoting angiogenesis and EMT. Additionally, studies have confirmed that IL-6 is upregulated in iCAFs, indicating that the expression of IL-6R in bladder cancer is responsive to IL-6 cytokine secreted by iCAFs. Because IL-6 blocking antibodies have been approved by the FDA for a variety of cancers, we added that this approach to interfering with CAF function in bladder cancer may be a clinically promising strategy. (page 6; lines 285-298)
Comments 6: Chapter 4 then covers the clinical trials. I wonder why these are separated from the description of the targets. Wouldnt it be easier to read and to comprehend if, after the original (and short) description of the molecular target, the information on therapy and clinical trials would follow?
Response 6: Thank you for your comment and I understand what you mean. When I wrote the paper, I also considered adding information about treatments and clinical trials after the description of the molecular target. However, after listing the generally known immune evasion factors in bladder cancer, we attempted to separately explain the targets and treatment strategies that could be linked to them. In your opinion, it is better to describe the molecular targets and clinical trials or therapeutic strategies one by one, but our view was to provide an overall analysis of the currently known targets and a comprehensive description of the possible strategies that could be proposed. Therefore, we have compiled a parallel list of treatment strategies worth utilizing. However, as you mentioned, the clinical trials of the candidates were highlighted in Chapter 3. (page 3; lines 144-154, page 6; lines 291-298) In the case of STn, DC is explained first among immune cells in Chapter 3, so it was revised to 4.6 accordingly.
Comments 7: However, this chapter also brings in new players: the chemokines, which have not been mentioned previously. That also appears rather arbitrary: why are they mentioned here, and not already in chapter 3, like most of the other targets? So, in conclusion, there is a lot of good content and its not difficult to read, but it couls still be significantly improved by focusing and making the entire story a more consistant narrative.
Response 7: Thank you for your comment, and I will explain that. Various immune cells contained within the TME actively participate in immune evasion responses and contribute to tumor growth. The function of each immune cell in bladder cancer was mainly covered in Chapter 3. And Chapter 4 presents treatment strategies applicable to the targets mentioned in Chapter 3 (ICI combinations, antagonists). Although the functions of immune cells are very diverse, we would like to suggest that it would ultimately be useful to block the chemokine mechanisms that are a prerequisite for their participation in TME formation. Rather than explaining the treatment strategy by dividing it by cell, I organized it like this because I wanted to emphasize the chemokine-targeted treatment strategy that affects each cell. But your comment pointed out something I hadn't thought of, and I appreciate that. So, we modified the title of Chapter 4 to link it to Chapter 3 rather than presenting the treatment strategy as a chemokine. (page 9; line 440)